# Microbiota-derived short chain fatty acids modulate microglia and promote Aβ plaque deposition

Alessio Vittorio Colombo[1], Rebecca Katie Sadler[2], Gemma Llovera[2], Vikramjeet Singh[2], Stefan Roth[2], Steffanie Heindl[2], Laura Sebastian Monasor[1], Aswin Verhoeven[3], Finn Peters[1], Samira Parhizkar[4], Frits Kamp[4], Mercedes Gomez de Aguero[5], Andrew J MacPherson[5], Edith Winkler[1,4], Jochen Herms[1,6,7], Corinne Benakis[2], Martin Dichgans[2,6], Harald Steiner[1,4], Martin Giera[3], Christian Haass[1,4,6], Sabina Tahirovic[1]†*, Arthur Liesz[2,6]†*

[1]German Center for Neurodegenerative Diseases (DZNE), Munich, Germany; [2]Institute for Stroke and Dementia Research (ISD), University Hospital, LMU Munich, Munich, Germany; [3]Center for Proteomics and Metabolomics, Leiden University Medical Center (LUMC), Leiden, Netherlands; [4]Metabolic Biochemistry, Biomedical Center (BMC), Faculty of Medicine, Ludwig-Maximilians-Universität München, Munich, Germany; [5]Maurice Müller Laboratories (DKF), Universitätsklinik für Viszerale Chirurgie und Medizin Inselspital, Bern, Switzerland; [6]Munich Cluster for Systems Neurology (SyNergy), Munich, Germany; [7]Center for Neuropathology and Prion Research, Ludwig-Maximilians University Munich, Munich, Germany

*For correspondence:
sabina.tahirovic@dzne.de (ST);
Arthur.Liesz@med.uni-muenchen.de (AL)

†These authors contributed equally to this work

**Abstract** Previous studies have identified a crucial role of the gut microbiome in modifying Alzheimer's disease (AD) progression. However, the mechanisms of microbiome–brain interaction in AD were so far unknown. Here, we identify microbiota-derived short chain fatty acids (SCFA) as microbial metabolites which promote Aβ deposition. Germ-free (GF) AD mice exhibit a substantially reduced Aβ plaque load and markedly reduced SCFA plasma concentrations; conversely, SCFA supplementation to GF AD mice increased the Aβ plaque load to levels of conventionally colonized (specific pathogen-free [SPF]) animals and SCFA supplementation to SPF mice even further exacerbated plaque load. This was accompanied by the pronounced alterations in microglial transcriptomic profile, including upregulation of ApoE. Despite increased microglial recruitment to Aβ plaques upon SCFA supplementation, microglia contained less intracellular Aβ. Taken together, our results demonstrate that microbiota-derived SCFA are critical mediators along the gut-brain axis which promote Aβ deposition likely via modulation of the microglial phenotype.

## Introduction

Alzheimer's disease (AD) is a progressive neurodegenerative disorder characterized by the aggregation and deposition of amyloid-β (Aβ) and tau. The identification of several AD risk genes such as TREM2, CD33, or CR1 as key regulators of microglial function triggered mechanistic studies revealing the contribution of the brain's resident innate immune cells to AD pathology. In particular, triggering microglial phagocytic clearance of Aβ can reduce amyloid plaque pathology (*Daria et al., 2017*; *Hansen et al., 2018*). Besides the already well-acknowledged contribution of amyloidogenic protein processing and neuroinflammation to AD pathology, the gut microbiome is emerging as a novel and highly relevant modifier of brain pathology. A key function of the gut microbiome has been established over the past decades in a number of neurological diseases spanning across

neurodevelopment, stroke, Parkinson's disease, and neuropsychiatric disorders (*Tremlett et al., 2017*). For example, we have previously described an intricate bidirectional link between the gut and brain after acute stroke, where stroke changes the gut microbiota composition (*Singh et al., 2016*). In turn, post-stroke dysbiosis induced changes in the immune response to stroke. Most importantly, the gut microbiota composition can be modulated to improve the disease outcome in stroke and other neurological disorders (*Benakis et al., 2020*), a finding that was robustly reproduced across disease entities and laboratories (*Cryan and O'Mahony, 2011*).

Only recently, a role of gut microbiota was also established in AD. AD patients have an altered gut microbiome compared to matched control patients, characterized by reduced species diversity and an increased abundance of Bacteroidetes (*Vogt et al., 2017*). Similar findings were obtained in 5xFAD and APPPS1 amyloidosis mouse models (*Brandscheid et al., 2017*; *Harach et al., 2017*). Other studies have identified that antibiotic treatment-induced changes in microbiota composition as well as microbial deficiency in germ-free (GF) animals were associated with reduced Aβ pathology (*Dodiya et al., 2019*; *Harach et al., 2017*; *Minter et al., 2017*; *Minter et al., 2016*). This effect could result from reduced Aβ production or increased clearance. Interestingly, GF animals as well as antibiotic-mediated dysbiotic animals exhibited reduced microglial activation (*Minter et al., 2017*). While previous studies documented a link between the gut microbiome and Aβ pathology, the underlying mechanisms and molecular mediators remain elusive. To address this key question, we generated a GF amyloidosis mouse model which allowed us to explore and identify bacterial metabolites that mediate gut-brain axis in AD. Moreover, we performed a detailed analysis of the gut microbiome's impact on amyloidogenesis and neuroinflammation, in order to differentiate between direct effects on Aβ generation and effects mediated via microglial cells. Our findings identify microbiota-derived short chain fatty acids (SCFA), bacterial fermentation products of fiber, as the sufficient mediator to promote Aβ plaque deposition. We further identified microglial transcriptomic and functional alterations that may underscore increased Aβ deposition triggered by SCFA.

## Results

### The gut microbiome promotes AD pathology

In order to study mechanisms of microbiota–brain interaction in AD, we generated a GF amyloidosis mouse model (APPPS1) by embryo transfer into axenic mice. GF APPPS1 mice were generated independent of previous studies using a similar approach (*Harach et al., 2017*; *Minter et al., 2016*). In accordance with these reports, we also observed a striking reduction in cerebral Aβ plaque load in 5 months old GF APPPS1 animals compared to their littermate GF APPPS1 animals which have been naturally recolonized (Rec) or APPPS1 mice housed under conventional, specific pathogen-free (SPF) conditions (*Figure 1A*). Although not powered to specifically investigate sex differences, an increase of plaque load by bacterial colonization was apparent also in a sex-specific analysis (*Figure 1—figure supplement 1*). Correspondingly, GF APPPS1 mice demonstrated a significantly better cognitive performance in a spatial memory task compared to recolonized and SPF mice (*Figure 1B*). We further analyzed the size distribution of Aβ plaques between the GF and SPF animals using an automated analysis paradigm of three-dimensional plaque segmentation and quantification in the frontal cortex (*Peters et al., 2018*) (*Figure 1C*). This showed that GF mice display a significantly decreased density of small (4–8 μm) Methoxy-X04-positive plaques compared to SPF, while larger plaques were not affected (*Figure 1D*). This result suggests that bacterial colonization might preferentially affect plaque formation rather than plaque growth.

### SCFA are bacterial metabolites contributing to AD pathology

As our initial experiments indicated a specific effect of the gut microbiome on the formation of small plaques, we focused our mechanistic studies on the early phase of Aβ plaque deposition, which corresponds to the 3 months of age in APPPS1 mouse model. Previous studies have demonstrated that SCFA (acetate, butyrate, and propionate) are metabolites produced by the gut microbiota which are critical mediators in various brain diseases such as stroke, Parkinson's disease, and neuropsychiatric disorders (*Dalile et al., 2019*). Therefore, we quantitatively analyzed the total plasma SCFA concentrations of acetate, butyrate, and propionate (C:2 C:4 SCFA) using gas chromatograph/mass spectrometry (GC/MS) based analysis in GF, recolonized and SPF mice. We observed an increase in

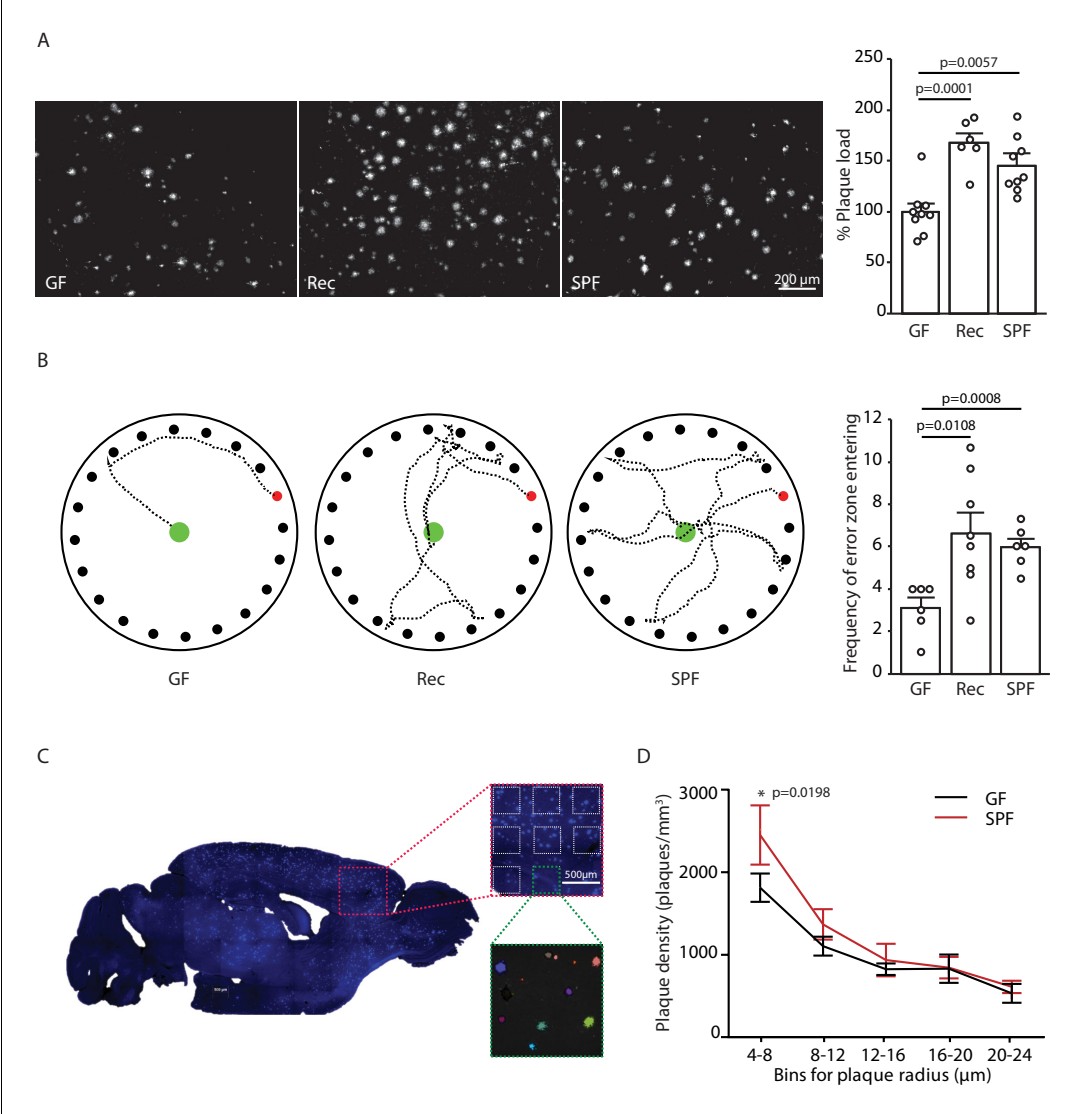

**Figure 1.** Germ-free APPPS1 mice show reduced Alzheimer's disease (AD) pathology. (**A**) Representative images and analysis of brain cortices from 5 months old germ-free (GF), naturally recolonized (Rec) and conventionally colonized (specific pathogen-free [SPF]) APPPS1 mice immunostained for Aβ (clone 6E10). Quantification of parenchymal plaque load reveals significantly reduced Aβ burden in GF mice compared to the Rec and SPF groups. Values are expressed as percentages of amyloid plaque area and normalized to GF group (ANOVA; n(GF) = 9, n(Rec) = 6, n(SPF) = 9). (**B**) Representative results and quantification from Barnes maze behavioral analysis of 5 months old GF, Rec and SPF APPPS1 mice. Quantification of frequency of error zone entering in the Barnes maze reveals a better performance in GF mice compared to Rec and SPF mice (ANOVA; n(GF) = 6, n(Rec) = 8, n(SPF) = 6). (**C**) Sagittal overview image indicating the analysis ROIs in the frontal cortex (blue: Methoxy-X04-positive plaques) and representative image demonstrating segmentation of Methoxy-X04 fluorescence intensity into individual plaques. The images show maximal intensity projections. Individual plaques are labeled with different colors. (**D**) Frequency distribution of plaque radius in 5 months old GF and SPF APPPS1 mice (two-way ANOVA; n(GF) = 5, n(SPF) = 5). All data are derived from at least three individual experiments and presented as mean ± SEM.

The online version of this article includes the following figure supplement(s) for figure 1:

**Figure supplement 1.** Sex-specific analysis of amyloid plaque load.

plasma concentrations for all three SCFA in SPF APPPS1 mice compared to GF conditions (*Figure 2A*). In order to study a causal link between SCFA and Aβ deposition, we performed a SCFA supplementation experiment by providing SCFA to GF APPPS1 mice. We used a combination of all three SCFA (*Figure 2B*), because all three were affected in plasma by bacterial colonization and this combination was also used by us and others in previous studies (*Erny et al., 2015*; *Sadler et al., 2020*). In accordance with previous reports, the used amounts of supplemented SCFA in the

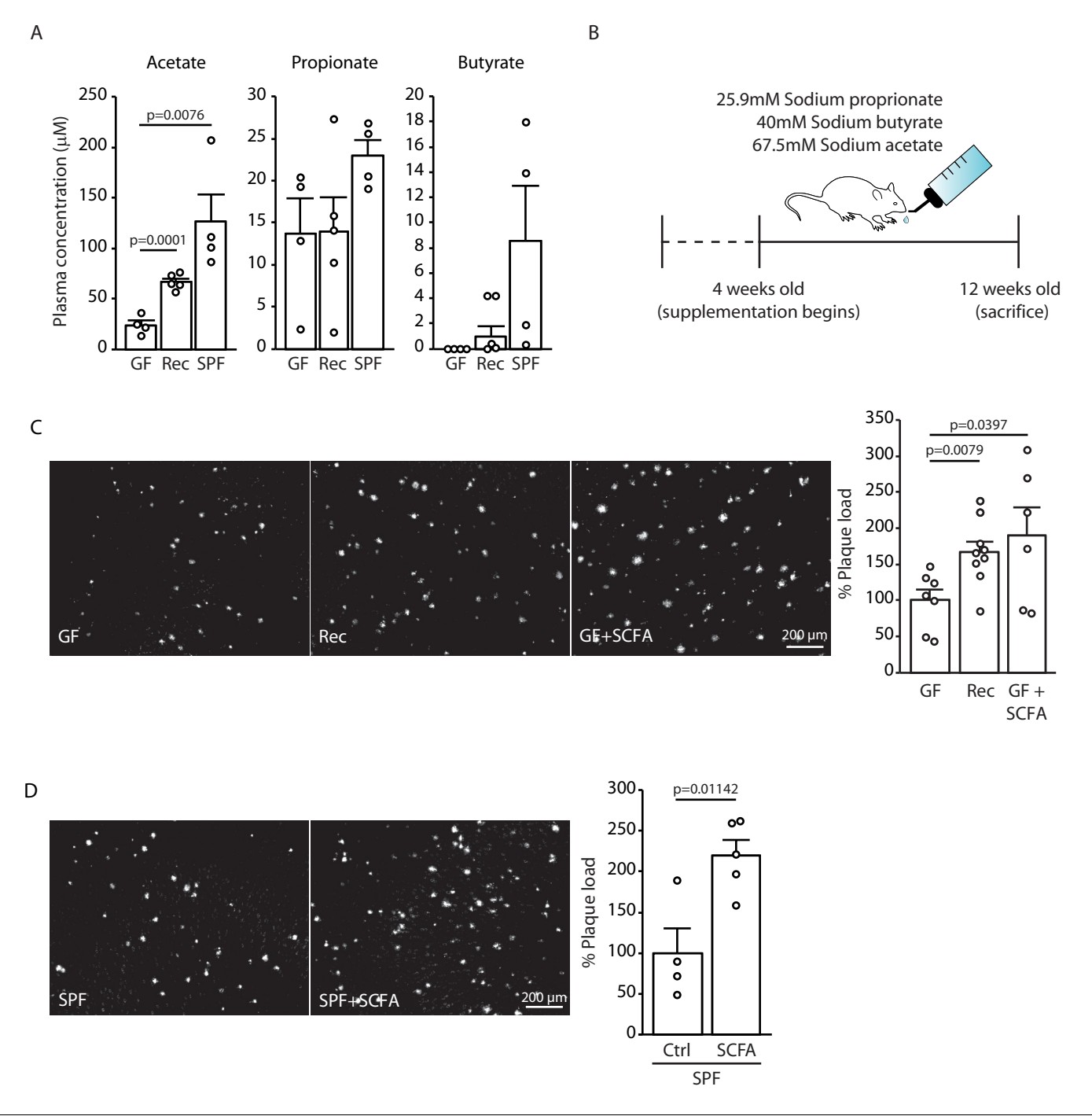

**Figure 2.** Short chain fatty acids (SCFA) are mediators of Aβ plaque deposition. (**A**) Plasma SCFA concentrations of acetate, butyrate, and propionate were quantified using GC/MS-based metabolomics analysis, showing an increase in specific pathogen-free (SPF) compared to germ-free (GF) mice (ANOVA; n(GF) = 4, n(Rec) = 5, n(SPF) = 4; two individual experiments). (**B**) Experimental plan for SCFA supplementation in GF mice. Four weeks old GF mice have been treated with SCFA in drinking water for 8 weeks. (**C**) Representative images from GF (control-treatment), Rec, and GF supplemented with SCFA mice showing a significant increase in Aβ plaque load upon SCFA administration (ANOVA; n(GF) = 7, n(Rec) = 9, n(GF +SCFA) = 6; three individual experiments). (**D**) SPF APPPS1 mice were supplemented with control or SCFA in drinking water for 4 weeks (from 8 to 12 weeks of age). Histological analysis revealed a significantly increased plaque load in SCFA-supplemented SPF mice compared to control treatment (unpaired T-test; n(Ctrl) = 4, n(SCFA) = 5; one individual experiment). All data in this figure are presented as mean ± SEM.

The online version of this article includes the following figure supplement(s) for figure 2:

**Figure supplement 1.** Total short chain fatty acids (SCFA) concentrations.

drinking water were sufficient to normalize plasma SCFA levels comparable to SPF mice (*Figure 2—figure supplement 1*). Strikingly, SCFA supplementation of GF APPPS1 mice was sufficient to nearly double cerebral Aβ plaque load (*Figure 2C*). Next, we treated SPF APPPS1 mice (4 weeks supplementation, starting at 8 weeks of age) in order to test a bona fide disease-promoting function of SCFA. Indeed, SCFA treatment of SPF mice significantly increased plaque load compared to control-treated SPF mice, revealing the potential of SCFA to worsen Aβ pathology in APPPS1 mice (*Figure 2D*). Taken together, our results demonstrate that SCFA are sufficient to mediate effects of bacterial gut colonization onto Aβ pathology.

## SCFA mildly increase amyloidogenic processing

After identifying SCFA as the key and sufficient mediator of the gut microbiome's effect on Aβ pathology, we aimed to identify the underlying mechanism. First, we explored potential direct effects of SCFA on amyloid precursor protein (APP) expression and processing by immunoblot analysis of brain tissue of GF APPPS1 mice with SCFA or control supplementation (*Figure 3A,B*). In accordance with the histological plaque load analysis, we identified markedly increased Aβ levels in SCFA- compared to control-supplemented GF APPPS1 mice. However, levels of the full-length APP (APP FL) were comparable between the groups. We observed only a mildly reduced ratio of the APP C-terminal fragment (CTFs) C83 (** in *Figure 3A*, produced by ADAM10 cleavage) and C99 (* in *Figure 3A*, produced by BACE1 cleavage). However, this slightly increased amyloidogenic APP processing upon SCFA supplementation to GF APPPS1 mice is unlikely to explain alone the substantially increased levels of Aβ (*Figure 3B*). We further analyzed protein levels of secretases responsible for APP processing (BACE1, ADAM10, and a catalytic subunit of γ-secretase PSEN1), but could not detect any alterations upon the SCFA supplementation of GF APPPS1 mice (*Figure 3A,B*). Moreover, using an in vitro γ-secretase activity assay (*Figure 3C*), we show that addition of SCFA does not change γ-secretase activity or processivity as reflected by comparable levels of total Aβ and unaltered profiles of Aβ species (Aβ37, 38, 40, and 42/43) in this cell free assay. In contrast to a previous study (*Ho et al., 2018*), we could exclude a direct effect of SCFA on Aβ aggregation (*Figure 3D*) as similar Aβ aggregation kinetics were determined in the presence or absence of SCFA in a Thioflavin T aggregation assay. Taken together, our data suggest that increased levels of Aβ triggered by SCFA in vivo are not mediated by major quantitative or qualitative changes in Aβ production but may rather be triggered by alterations in Aβ deposition and clearance.

## SCFA supplementation results in increased microglial activation

Given the critical role of microglia in AD and a demonstrated link between microglial function and Aβ pathology, we next focused on microglia as the potential cellular mediators of the SCFA effect on Aβ pathology (*Dodiya et al., 2019*). First, we observed that microglia in SCFA-supplemented GF APPPS1 mice had a significantly increased circularity index (CI; i.e. more amoeboid shape) indicating a more activated microglial phenotype (*Figure 4A*). We combined single molecule fluorescent in situ hybridization (smFISH, for microglial identification by *Cx3cr1* expression) and immunofluorescence (for plaque identification by staining with anti-Aβ 2D8 antibody) to visualize and quantify clustering of microglia around Aβ plaques. We observed increased microglial recruitment to Aβ plaques in SCFA- compared to control-treated APPPS1 GF mice (*Figure 4B*). Next, we investigated the influence of bacterial colonization on microglial reactivity in the WT background. To this end, we injected brain homogenates from 8 months old APPPS1 mice containing abundant Aβ into the hippocampus of GF or SPF WT mice (*Figure 4C*) and subsequently analyzed microglial abundance and *Trem2* mRNA expression as a marker of microglial activation by smFISH. We observed a significant increase in overall microglial cell counts at the peri-injection site of SPF compared to GF WT mice (*Figure 4D*). Moreover, microglia in SPF mice expressed significantly more *Trem2* mRNA puncta per microglia compared to GF WT mice (*Figure 4E*). Next, we questioned whether this SCFA-induced change in microglial recruitment and reactivity might also be associated with an altered phagocytic capacity of microglia, which is one of the most intensively investigated microglial functions in the context of AD. Therefore, we assessed the recruitment of microglia to Aβ plaques and their phagocytic uptake. This was analyzed by SCFA- or control-treatment in SPF APPPS1 mice to avoid potential counteracting effects of GF mice on microglial recruitment or phagocytosis. Corresponding to the increased recruitment and reactivity in GF SCFA-supplemented mice, we also observed a

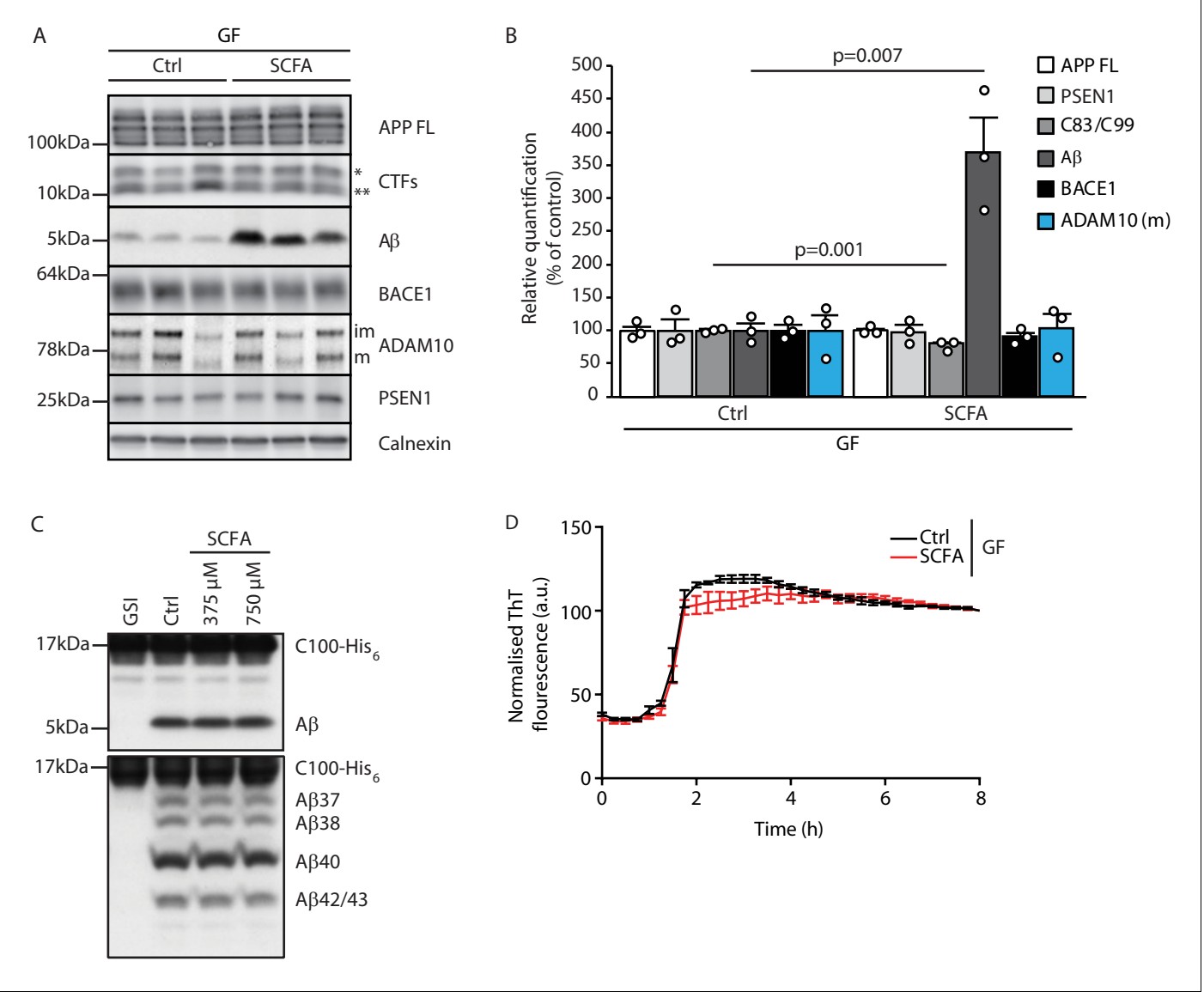

**Figure 3.** Short chain fatty acids (SCFA) mildly increase amyloidogenic processing. (**A**) Western blot analysis and (**B**) its densitometry quantification of 3 months old brain homogenates of control (Ctrl)- and SCFA-supplemented germ-free (GF) APPPS1 animals. The Aβ level is significantly increased in SCFA group in comparison to Ctrl, despite unaffected APP FL levels. APP CTFs show a decreased C83 (**) to C99 (*) ratio. We could not detect alterations in protein levels of secretases involved in APP processing (ADAM10, BACE1, and γ-secretase/PSEN1). m = ADAM10 mature form; im = ADAM10 immature form. Data represent mean ± SEM (unpaired T-test; n(Ctrl) = 3, n(SCFA) = 3). (**C**) Upper panel: γ-Secretase reconstituted into lipid vesicles was incubated at 37°C together with the C99-based substrate C100-His$_6$ in the presence of increasing doses of a SCFA mixture (375 and 750 µM final concentration of total SCFA of an equimolar mixture of Na-acetate, Na-propionate, and Na-butyrate) for 24 hr. Production of Aβ was analyzed by immunoblotting. γ-Secretase inhibitor (GSI) L-685,458 (0.4 µM) was used as a negative control. No alterations in Aβ levels were detected in the presence of SCFA. Lower panel: Qualitative analysis of individual Aβ species via Tris-Bicine-Urea SDS-PAGE reveals that SCFA treatment does not alter the ratio among the different Aβ species (Aβ37-38-40-42/43) suggesting no direct effects on modulation of γ-secretase cleavage. (**D**) Aggregation kinetics of monomeric Aβ40 recorded by the increase in fluorescence of Thioflavin T incubated with either 30 mM NaCl (Ctrl) or 30 mM SCFA mixture do not show any significant difference, suggesting that SCFA do not directly modify Aβ fibrillarization. Data points represent mean ± SD from three independent experiments.

significantly increased microglial recruitment to Aβ plaques in SCFA-supplemented SPF APPPS1 mice (*Figure 4F*). In contrast to control-treated mice, we detected significantly less microglia with intracellular Aβ in SCFA-treated mice that is in accordance with the increased Aβ pathology (*Figure 4F* right). However, when using an ex vivo amyloid plaque clearance assay, we did not

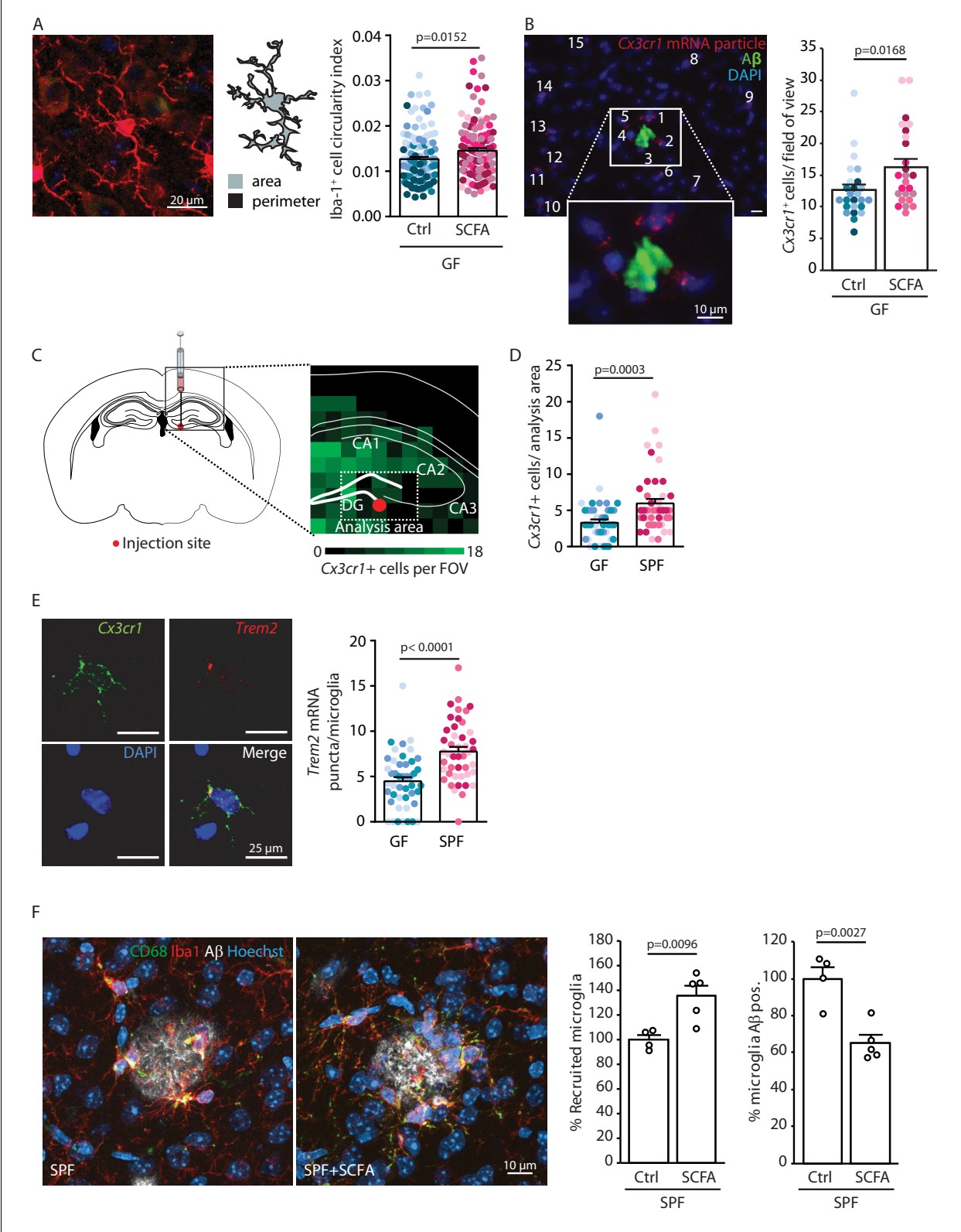

**Figure 4.** Short chain fatty acids (SCFA) modulate microglia. (**A**) Morphological analysis of microglia shows an increase in the circularity index, indicating a more activated phenotype, in SCFA- compared to control-treated germ-free (GF) APPPS1 mice. Iba1 (red) has been used as microglial marker. For each group, each shade of color represents the microglia from a single mouse (U test; n(Ctrl) = 6, n(SCFA) = 5). (**B**) smFISH analysis of microglial cells (red, *Cx3cr1* mRNA particles) surrounding amyloid plaques (green, anti-Aβ clone 2D8) shows an increased number of *Cx3cr1*-positive cells (>4 puncta)

*Figure 4 continued on next page*

Figure 4 continued

clustering around Aβ plaques in SCFA- compared to control-supplemented GF APPPS1 mice. DAPI (blue) was used as nuclear dye (U test; n(Ctrl) = 5, n (SCFA) = 5 mice, 5 FOV per mouse). (C) Experimental outline for APPPS1 brain homogenate injection into a WT mouse brain showing the injection site (red dot). (D) smFISH analysis of *Cx3cr1*-positive cells (>4 puncta) surrounding the APPPS1 brain homogenate injection site (analysis area relative to injection site for all brains) showing an enhanced recruitment of microglial cells in specific pathogen-free (SPF) versus GF WT mice. (E) APPPS1 brain homogenate injection induces higher microglial activation in SPF in comparison to GF WT mice as shown by the higher amount of *Trem2* mRNA puncta (red) per *Cx3cr*1-positive (green) microglia. DAPI (blue) was used as nuclear dye. In D and E, each different shade of color represents the microglia from a single mouse (U test, n(GF) = 3, n(SPF) = 3; two individual experiments, 120 analyzed images (with multiple microglia) per mouse). (F) Microglial recruitment and Aβ plaque uptake were histologically quantified in SCFA- and control-supplemented SPF APPPS1 mice. Although significantly more CD68-positive (green) and Iba1-positive (red) microglial cells were located at Aβ plaques (white, anti-Aβ clone 3552), the number of Aβ-positive microglia was significantly reduced in the SCFA- compared to the control-treated group. Values are expressed as percentages and normalized to the control-treated group (unpaired T-test; n(Ctrl) = 4, n(SCFA) = 5; one individual experiment).

The online version of this article includes the following figure supplement(s) for figure 4:

**Figure supplement 1.** Ex vivo amyloid plaque clearance assay.

detect a direct effect of SCFA-treatment on the phagocytic capacity of microglia (*Figure 4—figure supplement 1*).

## Microglia-derived ApoE expression is increased upon supplementation by SCFA

To analyze the effect of SCFA on microglia in more detail, we performed a transcriptome Nanostring analysis of brain samples from control- and SCFA-supplemented GF APPPS1 mice at 3 months of age. Corresponding to the results of the smFISH and histological analysis (*Figure 4A,B*), we found numerous candidate genes previously associated with microglial activation to be upregulated in SCFA-supplemented APPPS1 animals (*Figure 5A,B*). Many of the most abundantly regulated microglial genes were associated with secretory functions (chemokines and complement factor secretion) and with pathogen recognition (e.g. *Tlr7, Trem2, Tyrobp,* and *Myd88*) (*Figure 5B*). A biological network analysis revealed upregulation of ApoE and subsequent activation of the ApoE-TREM2 pathway as a central biological pathway effected by SCFA (*Figure 5C*). In accordance with the transcriptomic data, we further confirmed an increase in ApoE protein expression in SCFA- compared to control-supplemented GF APPPS1 mice by immunohistochemistry (*Figure 5D*). Previous studies have indicated astrocytes and microglia as major sources of cerebral ApoE (*Holtzman et al., 2012*; *Shi and Holtzman, 2018*). Therefore, we analyzed by immunohistochemistry the amounts of ApoE protein in control- and SCFA-supplemented GF APPPS1 mice and detected an increase of ApoE coverage colocalizing with microglial cells but not with GFAP-positive astrocytes. Correspondingly to bacterial colonization at 5 months of age (*Figure 1D*), also SCFA treatment of 3 months old APPPS1 mice resulted specifically in an increase of small plaques (4–8 µm radius) (*Figure 5F*).

A direct effect of SCFA on the microglial transcriptomic profile was further confirmed by treating cultured primary microglia from SPF WT mice with SCFA. As expected and previously described, the overall transcriptomic profile between microglia in vitro differed from the in vivo profile, as also illustrated by the reduced ApoE expression (*Bohlen et al., 2017*; *Butovsky et al., 2014*). Nevertheless, the combined SCFA treatment induced a pronounced upregulation of genes previously associated with inflammatory functions of microglia (*Figure 5—figure supplement 1A*). Next, using this model system, we compared the impact of individual SCFA (acetate, butyrate, and propionate) and the combined treatment on the microglial transcriptome. Analysis of individual genes as well as biological processes by pathway analysis revealed a more abundant overlap of butyrate and propionate than acetate with the combined SCFA treatment effect (*Figure 5—figure supplement 1B,C*). Yet, none of the individual SCFA effects were sufficient to achieve the changes in the transcriptomic profile of the combined SCFA treatment, suggesting a synergistic effect of single SCFA on modulating microglia.

## Discussion

Recent studies have demonstrated a key role of the gut microbiome in Aβ pathology in AD. However, the mechanisms by which the microbiome modulates disease progression and the molecular

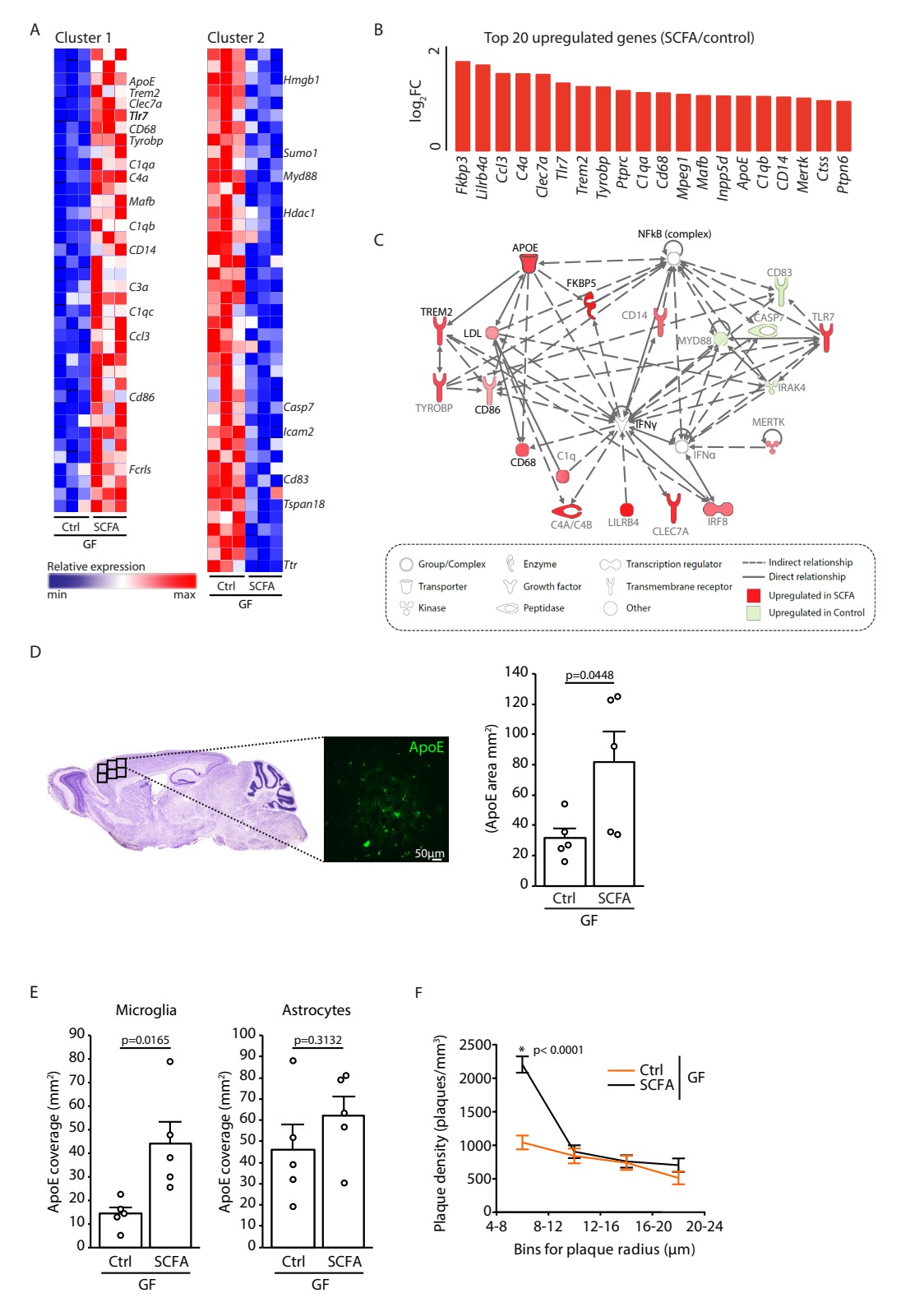

**Figure 5.** Increased ApoE expression marks microglial activation upon short chain fatty acids (SCFA) supplementation. (**A**) Heatmap of Nanostring transcriptomic analysis from control- and SCFA-supplemented germ-free (GF) APPPS1 mice at 3 months of age. Row values were scaled using unit variance scaling. Genes previously associated with microglial function have been annotated on the heatmap. Three mice per group have been analyzed. (**B**) Top 20 upregulated genes in SCFA- *versus* control-treated samples. Most of transcriptome hits have been previously associated with

*Figure 5 continued on next page*

*Figure 5 continued*

microglial activation. (**C**) Functional gene interaction network analysis using Ingenuity Pathway Analysis. Genes are colored based on fold-change values determined by RNA-Seq analysis, where red indicates an increase in SCFA- and green in control-treated animals. Network analysis revealed upregulation of the ApoE-TREM2 axis as one of the principal biological pathways upregulated by SCFA. (**D**) Representative sagittal brain section indicating location of the analyzed region of interest in the frontal cortex and representative image showing ApoE (green) distribution. Quantification of ApoE signal showed a SCFA-dependent increase of ApoE expression (unpaired T-test; n(Ctrl) = 5, n(SCFA) = 5; three individual experiments). (**E**) Quantification of ApoE colocalization (absolute coverage area in mm$^2$) with microglia and astrocytes in control- and SCFA-supplemented GF APPPS1 mice at 3 months of age (unpaired T-test; n(Ctrl) = 5, n(SCFA) = 5; two individual experiments). (**F**) Analysis of Methoxy-X04-stained brain sections showed a specific increase in plaques of smaller sizes (4–8 μm radius) in 3 months old SCFA- compared to Ctrl-supplemented GF APPPS1 mice (unpaired T-test per bin; n(Ctrl) = 5, n(SCFA) = 5; three individual experiments). Data represent mean ± SEM.

The online version of this article includes the following figure supplement(s) for figure 5:

**Figure supplement 1.** Effect of individual short chain fatty acids (SCFA) on microglial polarization.

mediators along the gut-brain axis have remained unknown. In this study, we identified microbiota-derived SCFA as microbial metabolites contributing to Aβ plaque deposition in the brain. GF APPPS1 mice have markedly reduced SCFA concentrations and reduced plaque load, and supplementation with SCFA is sufficient to mimic the microbiome's effect and increase Aβ plaque burden. We identified microglia as the key cell population responsive to SCFA and increased microglial ApoE may mediate accelerated Aβ deposition during early stages of amyloidogenesis.

The gut bacteria have an intricate function in preserving the integrity of an intestinal–epithelial barrier, but exceeding this function, they have also been demonstrated to be critical for shaping the immune system, host metabolism, and nutrient processing (*Bäckhed et al., 2004*; *Hooper et al., 2001*; *Round et al., 2011*). Indeed, the functions of the gut microbiome extend beyond the intestinal tract. With their rich repertoire of antigens, metabolites, and direct interaction with the autonomic nervous system the microbiome potently influences remote organ function including the brain. The microbiome plays an important role in microglial maturation during development and in adulthood (*Abdel-Haq et al., 2019*). Previously believed to be shielded from the peripheral immune system and blood-borne mediators, it became apparent that the brain is accessible for microbial metabolites to affect cerebral immunity in health and disease (*Abdel-Haq et al., 2019*; *Janakiraman and Krishnamoorthy, 2018*).

In accordance with our observation of SCFA to be the mediator along the gut-brain axis in AD, multiple lines of evidence have indicated that SCFA might be the key microbial metabolite group acting on brain function. Particularly, SCFA have been demonstrated to induce microglial maturation using a similar analysis paradigm by SCFA supplementation in GF mice as used in our study (*Erny et al., 2015*). In a Parkinson's disease mouse model, SCFA were sufficient to induce neuroinflammation and disease progression in the absence of the gut microbiome (*Sampson et al., 2016*). Moreover, SCFA have been implicated in a wide range of brain disorders or physiological functions under cerebral control such as mood disorders, autism spectrum disease, or energy metabolism (*Cryan et al., 2019*; *Li et al., 2018*; *van de Wouw et al., 2018*). Specifically, for AD patients it has previously been shown that they have alterations in microbiota composition (*Haran et al., 2019*; *Vogt et al., 2017*). Thus, it is conceivable that SCFA concentrations might be altered in AD patients (*Nagpal et al., 2019*). One previous clinical study focusing on the association between SCFA concentration and AD disease burden suggested a correlation between amyloid load and blood SCFA concentrations (*Marizzoni et al., 2020*). Yet, all these clinical evidence of a potential association between SCFA concentrations and disease progression in AD is rather preliminary and requires confirmation in prospectively recruited and larger cohorts.

Over the past decade, microglial cells have come into the focus of experimental AD research. This has been driven by the identification of several genetic AD risk loci being associated with microglial cells (*Long and Holtzman, 2019*). Moreover, microglia have been demonstrated in murine AD models to actively contribute to various pathophysiological process in AD such as plaque seeding, plaque phagocytosis, and neuronal dysfunction by synaptic pruning (*Heneka et al., 2014*; *Heneka et al., 2013*; *Hong et al., 2016*; *Sarlus and Heneka, 2017*). Previous proof-of-concept studies investigating a potential role of the microbiome in AD have already indicated an effect of microbial colonization or antibiotic treatment on microglial activation (*Dodiya et al., 2019*; *Harach et al., 2017*; *Minter et al., 2017*). Although previous studies did not reveal a mechanistic link between

microbiome function and Aβ plaque pathology, they have reproducibly shown that microbiota eradication (GF mice) or impairing the microbiome's metabolic function (antibiotic treatment) reduced microglial activation and Aβ plaque load, which is in accordance with our study. This may at first glance appear contradicting with the research hypothesis supporting microglial activation and their corresponding increased phagocytic clearance as an approach to reduce Aβ plaque load (*Daria et al., 2017*; *Guillot-Sestier et al., 2015*). However, we should bear in mind a potentially dual nature of microglial responses and their concomitant beneficial and detrimental roles in AD (*Lewcock et al., 2020*). Accordingly, previous studies have unequivocally demonstrated a role of microglia in promoting plaque seeding and growth, particularly at early stages of AD pathology. For example, microglia have been attributed to promote plaque seeding by the release of pro-inflammatory protein complexes, the so-called ASC specks (*Venegas et al., 2017*). A recent elegant study using sustained microglia depletion starting before plaque pathogenesis revealed a critical role of microglia in the formation of Aβ plaques and plaque density (*Spangenberg et al., 2019*). On the other hand, preventing microglial activation, such as upon loss of Trem2, increased amyloid seeding due to reduced phagocytic clearance of Aβ seeds, but at the same time reduced plaque-associated ApoE (*Parhizkar et al., 2019*). Previous studies demonstrated that ApoE co-aggregates with Aβ fibrils and contributes to plaque seeding and plaque core stabilization (*Liao et al., 2015*), thus underscoring the complexity of microglia–ApoE interactions during amyloidogenesis.

ApoE is predominantly produced by astrocytes under physiological conditions, but upregulated in AD microglia (*Sala Frigerio et al., 2019*; *Sebastian Monasor et al., 2020*). Indeed, microglial depletion in the 5xFAD mouse model resulted in a marked reduction of plaque-associated ApoE (*Spangenberg et al., 2019*). In the present study we identified upregulation of microglial ApoE expression. We hypothesize that increased ApoE expression may contribute to SCFA-driven microglial activation and increased plaque load. However, final proof for this link requires investigation of SCFA effects in mice with microglia-specific ApoE depletion and awaits future studies.

SCFA use membrane-receptors as well as receptor-independent mechanisms to enter target cells and exert their functions. Receptors specifically activated by SCFA (free fatty-acid receptor-2 and -3) have been identified in several organs, including the brain (*Layden et al., 2013*). Yet, the SCFA can also freely enter target cells by simple diffusion or fatty acid transporters and exert their biological function intracellularly, independent of specific receptors (*Lin et al., 2012*; *Moschen et al., 2012*). Therefore, the most likely target mechanism to interfere with SCFA effects in microglia could be on the level of chromatin remodeling and regulation of target gene expression as it has previously been demonstrated (*Davie, 2003*; *Huuskonen et al., 2004*). However, the detailed mechanistic understanding of SCFA on microglial transcriptomic and functional regulation remains unclear and requires future investigations.

Besides direct effects on brain function including glial cell homeostasis, SCFA can also affect peripheral immune cells and thereby indirectly modulate neuroinflammatory mechanisms (*Cryan et al., 2019*). SCFA have been demonstrated to be potent immune modulators particularly of T cell function and their polarization into pro- and anti-inflammatory subpopulations (*Smith et al., 2013*). In fact, we have previously demonstrated that the effect of SCFA on post-stroke recovery in an experimental brain ischemia model was dependent on circulating T cells (*Sadler et al., 2020*; *Sadler et al., 2017*). In these previous studies, circulating T cells mediated the effects of SCFA on the cerebral micromilieu either by their reduction in cerebral invasion or by polarization of the secreted cytokine profile. However, in contrast to AD, stroke induces a pronounced neuroinflammatory response to the acute tissue injury with the invasion of large numbers of circulating T cells, which have been demonstrated to contribute substantially to stroke pathology (*Cramer et al., 2019*; *Liesz et al., 2009*). Therefore, while the peripheral polarization of T cells and their consecutive impact on local neuroinflammation is conceivable after stroke, this pathway seems less likely in AD with only very limited invasion of circulating immune cells into the brain, particularly in the early stages of AD pathology. Moreover, the effect of SCFA on the peripheral immune compartment is most likely site-dependent and might occur within the intestinal wall, blood circulation of secondary lymphatic organs. It is a limitation of this study that SCFA supplementation was performed only by oral administration (in drinking water) which might lead to unphysiologically high SCFA concentrations in the upper gastrointestinal tract while physiological SCFA as fermentation products of the microbiota occurs predominantly in the colon (*Morrison and Preston, 2016*). While site-specific

effects of SCFA and differing resorption routes were beyond the scope of the present proof-of-concept study, these aspects are critical for the clinical translation and require further studies.

Interestingly, the biological effect of SCFA on microglia seems to be largely dependent on the specific disease condition. While SCFA in neurodegenerative conditions, as also in our study, have been associated with microglial reactivity and activation, their function in primary autoimmune and acute brain disorders has mainly been described to be anti-inflammatory. For example, we have been previously shown that SCFA treatment promotes anti-inflammatory mechanisms and improve neuronal function in ischemic stroke and similar findings have also been reported in experimental autoimmune encephalitis (*Haghikia et al., 2015*; *Sadler et al., 2020*). The detailed cause of this divergent functions of SCFA in different disease conditions is currently still unknown and requires further exploration for the potential use of personalized treatment approaches. Yet, in light of our results in the AD compared to the stroke model, it seems likely that the different effects of SCFA treatment on disease outcome could be due to engaging different mechanistic routes, such as peripheral lymphocyte polarization in EAE and stroke versus local effects on microglia in AD.

Our study suggests that SCFA-regulated pathways might be promising drug targets in the peripheral circulation for early-stage AD to prevent microglial activation, ApoE production, and the development of amyloid pathology. However, the therapeutic targeting and neutralization of SCFA, e.g. by specific SCFA-scavengers, in order to chronically reduce circulating SCFA blood concentrations is currently not established. Also attempting to reduce SCFA concentrations by reduction of nutritional fiber intake is not a feasible therapeutic approach. Reduced fiber intake correlates with increased risk of metabolic syndrome and cardiovascular events such as myocardial infarction and stroke. Furthermore, dietary fiber restriction will most likely not be efficient to affect microglial activation in early-stage AD because a near-complete reduction of blood SCFA concentrations, as seen in GF mice, will not be achieved by dietary intervention.

In conclusion, our study identifies SCFA as molecular mediators along the gut-brain axis in AD. Identification of this novel pathway will open up new avenues for therapeutic targeting of the microbiome-SCFA-microglia axis to reduce the inflammatory impact on AD development.

# Materials and methods

## Key resources table

| Reagent type (species) or resource | Designation | Source or reference | Identifiers | Additional information |
|---|---|---|---|---|
| Genetic reagent (*Mus musculus*) | APPPS1 | doi: 10.1038/sj.embor.7400784 | MGI:3765351 | C57BL/6 background |
| Antibody | Anti-IBA1 (guinea pig polyclonal) | SYSY | Cat# 234 004, RRID:AB_2493179 | IHC (1:500) |
| Antibody | Anti-IBA1 (rabbit polyclonal) | Wako | Cat#:019–19741, RRID:AB_839504 | IHC (1:500) |
| Antibody | Anti-CD68 (rat monoclonal) | Bio-Rad | Cat#:MCA1957G, RRID:AB_324217 | IHC (1:500), WB (1:1000) |
| Antibody | Anti-Amyloid Y188 (rabbit monoclonal) | Abcam | Cat#:1565–1, RRID:AB_562042 | WB (1:1000) |
| Antibody | Anti-Amyloid beta 3552 (rabbit polyclonal) | doi: 10.1523/JNEUROSCI.5354–05.2006 | | IHC (1:500), WB (1:2000) |
| Antibody | anti-beta-Amyloid, 6E10 (mouse monoclonal) | BioLegend | Cat# 803002, RRID:AB_2564654 | IHC (1:500) |
| Antibody | Anti-Presenilin 1 (NT1) (mouse monoclonal) | BioLegend | Cat# SIG-39194–500, RRID:AB_10720504 | WB (1:1000) |
| Antibody | Anti-Human Adam10 (mouse monoclonal) | R and D Systems | Cat# MAB1427, RRID:AB_2223057 | WB (1:1000) |
| Antibody | Anti-BACE1 (rabbit monoclonal) | Epitomics | Cat# 2882–1, RRID:AB_2061494 | WB (1:1000) |
| Antibody | Anti-Calnexin (rabbit monoclonal) | Stressgen | Cat# ADI-SPA-860, RRID:AB_10616095 | WB (1:1000) |

*Continued on next page*

*Continued*

| Reagent type (species) or resource | Designation | Source or reference | Identifiers | Additional information |
|---|---|---|---|---|
| Chemical compound, drug | Sodium butyrate 98% | Sigma Aldrich | 303410 | In vitro: 250 µM<br>In vivo: 40 mM |
| Chemical compound, drug | Sodium propionate ≥ 99.0% | Sigma Aldrich | P1880 | In vitro: 250 µM<br>In vivo: 25.9 mM |
| Chemical compound, drug | Sodium acetate anhydrous ≥ 99% | Sigma Aldrich | S2889 | In vitro: 250 µM<br>In vivo: 67.5 mM |
| Antibody | Anti-Amyloid beta 2D8 (rat polyclonal) | DOI: 10.1016/j.nbd.2007.04.011 | | WB (1:1000)<br>IHC (1:300) |
| Antibody | Anti-P2Y12 (rabbit polyclonal) | AnaSpec | Cat# 55043A, RRID:AB_2298886 | IHC (1:100) |
| Antibody | Anti-GFAP (rabbit polyclonal) | Dako | Cat# Z0334, RRID:AB_10013382 | IHC (1:200) |
| Antibody | Anti-APOE (mouse monoclonal) | DOI 10.1172/JCI96429 | Clone HJ6.3B | IHC (1:50) |
| Chemical compound, drug | Thiazine red | VWR Chemicals | 27419.123 | IHC (2 µM) |
| Chemical compound, drug | Hoechst 33342 | ThermoFisher | H3570 | IHC (1:2000) |

## Animal experiments

All animal experiments were performed under the institutional guidelines for the use of animals for research and were approved by the governmental ethics committee of Upper Bavaria (Regierungspraesidium Oberbayern, license number #160–14). SPF B6.Cg-Tg (Thy1-APPSw,Thy1-PSEN1*L166P) 21Jckr (APPPS1) (*Radde et al., 2006*) mice were bred for this project at the core animal facility of the Center for Stroke and Dementia Research in Munich. SPF mice in this study were kept at the core animal facility of the Center for Stroke Dementia Research (Munich, Germany) in individually ventilated cage systems at 12 hr dark–light cycle with ad libitum access to food and water.

## GF mouse generation and handling

APPPS1 mice were rederived to GF status in the Clean Mouse Facility, University of Bern, Switzerland as previously reported (*Harach et al., 2017*) and housed in flexible-film isolators. All mouse handling and cage changes were performed under sterile conditions. GF and SPF mice all received the same autoclaved chow and sterile water. For SCFA treatment, mice were given a sterile-filtered solution containing 25.9 mM sodium propionate, 40 mM sodium butyrate, and 67.5 mM sodium acetate in sterile water ad libitum starting from 4 (treatment of GF mice) or 8 (treatment of SPF mice) until 12 weeks of age. The SCFA water solution was renewed every 3 days. For surgical interventions (stereotactic injection, see below) in GF mice, the whole surgical procedure and post-surgical care was performed in a microbiological safety cabinet as previously described in detail (*Singh et al., 2018*). Animals were regularly checked for germ-free status by aerobic and anaerobic cultures, cecal DNA fluorescence stain, and 16 s rRNA PCR of fecal pellets. GF status of all animals used in this study (control and SCFA-supplemented) has been confirmed after sacrificing the mice.

## Natural recolonization of GF mice

GF littermate mice were naturally colonized through co-housing with conventional SPF mice from the same SPF animal facility in which the conventional SPF APPPS1 mice were housed (Center for Stroke and Dementia Research, Munich). Co-housing was started at 4 weeks of age and maintained until sacrifice of the animals at 3 months of age. Animals were housed in HAN-gnotocage mini-isolators and received sterile food pellets and water as GF mice.

## Histological analysis of Aβ plaque load and density

Mice were transcardially perfused with PBS followed by overnight post fixation with 4% PFA solution. Free floating 30 µm sagittal brain sections have been permeabilized and blocked for 1 hr in PBS/0.5% Triton x-100/5% normal goat serum (NGS). Next, samples have been incubated overnight

at 4°C with primary antibody anti β-amyloid (Aβ) (clone 6E10, 1:500, BioLegend) diluted in blocking buffer and stained with the corresponding goat secondary antibody. Immunostainings have been performed on six brain sections/animal collected every 300 µm starting from the interhemispheric fissure. Three 10× images/section (front, middle, and rear cortex) have been acquired and the plaque load (6E10 coverage area) has been determined using the particle analysis tool in ImageJ software (NIH) and normalized on total tissue area. Analysis has been performed at least on five mice per group/time point. Histological analysis has been performed by an investigator (AVC) blinded to treatment groups. For analysis of plaque density, 30 µm sections were stained with Methoxy-X04 and confocal images were collected in Z-stacks. Eight ROI were selected across the cortex and acquired across the different cortical layers. Image data analysis was performed as previously reported in detail (*Peters et al., 2018*). In brief, local background correction was applied to diminish intensity variations among different stacks and to account for the intensity decline in the axial dimension due to absorption and scattering of photons. For this purpose, the voxel intensity was normalized in each Z-layer to the 70th percentile of Methoxy-X04 fluorescence intensity. Subsequently, amyloid plaques were defined by applying the 90th percentile in the Methoxy-X04 fluorescence data stacks. The radius of each individual plaque was calculated from the Z-plane with the largest area extension in XY (radius=$\sqrt{(area/\pi)}$), assuming spherical shape of plaques (*Hefendehl et al., 2011*). All plaques that contacted the image border were excluded from the analysis. The cut-off size was set to a minimal plaque radius of 2 µm.

## Barnes maze test for memory deficits

A modified version of *Attar et al., 2013* was used to perform the Barnes maze test. The elevated 20-hole apparatus (diameter: 100 cm, hole diameter: 10 cm) had a target box which was placed under the maze. The protocol includes three phases of interaction of mice with the maze: (1) habituation to the maze, (2) a 2-day training period, and (3) a probe trial 48 hr later. Before each day of training or probe mice were placed 30 min prior procedure in the testing room for acclimatization. On day 1 mice were habituated to the maze. Therefore, mice were placed in the center of the maze in a 2 l glass beaker. After 1 min of acclimatization, mice were guided slowly by moving the glass beaker toward the target hole. This was done three consecutive times per mouse. On day 2 mice were placed in center and given the possibility to freely explore and find the target hole. If mice did not reach the target hole within 3 min the glass beaker was used to slowly guide them to the target. This was done in three consecutive trials. On day 3 mice were placed under the beaker and 10 s after placement the beaker was removed, and mice were allowed to explore freely and find the target hole. Again, three consecutive trials were performed with every mouse. This procedure was repeated for the actual probe 48 hr later. Acquisition and zone-dependent analysis were performed with Ethovision XT (Noldus). Data acquisition and analysis were performed by an investigator (SR) blinded to treatment groups.

## GC/MS-based SCFA analysis

The SCFA acetate, propionate, and butyrate were quantified as described before (*Hoving et al., 2018*). Briefly, 10 µl of plasma was transferred to a glass vial containing 250 µl acetone (Sigma-Aldrich), 10 µl one ppm internal standards solution containing acetic acid-d4, propionic acid-6, and butyric acid-d8 (Sigma Aldrich), and 10 µl ethanol. Thereafter, samples were derivatized with pentafluorobenzyl bromide (PFBBr), as follows: 100 µl 172 mM PFBBr (Thermo) in acetone was added, samples were mixed and heated to 60°C for 30 min. After the samples had cooled down to room temperature a liquid–liquid extraction was performed using 500 µl n-hexane (Sigma-Aldrich) and 250 µl LC-MS grade water. The upper n-hexane layer was transferred to a fresh glass vial and subsequently used for GC-MS analysis. Calibration standards were prepared analogous. For calibration standards no plasma was added and 10 µl of EtOH was replaced by 10 µl standards solution (Sigma-Aldrich) in EtOH. Samples were analyzed on a Bruker Scion 436 GC fitted with an Agilent VF-5ms capillary column (25 m × 0.25 mm i.d., 0.25 µm film thickness) coupled with a Bruker Scion TQ MS. Injection was performed using a CTC PAL autosampler (G6501-CTC): 1 µl sample was injected splitless at 280°C. Helium 99.9990% was used as carrier gas at a constant flow of 1.20 ml/min. The GC temperature program was set as follows: 1 min constant at 50°C, then linear increase at 40°C/min to 60°C, kept constant for 3 min, followed by a linear increase at 25°C/min to 200°C, linearly increased

at 40°C/min to 315°C, kept constant for 2 min. The transfer line and ionization source temperature were 280°C. The pressure of the chemical ionization gas, methane (99.9990%), was set at 15 psi. Negatively charged ions were detected in the selected ion monitoring mode, and acetic acid, acetic acid-d4, propionic acid, propionic acid-d6, butyric acid, and butyric acid-d8 were monitored at $m/z$ 59, 62, 73, 78, 87, and 94 respectively.

## Sample preparation for biochemical analyses

Mouse brains were isolated and immediately snap frozen in liquid nitrogen. Frozen brains were then mechanically pulverized for further applications and stored at −80°C. Aliquots of brain powder were lysed for 20 min in lysis buffer (150 mM NaCl, 50 mM Tris pH 7.5, 1% Triton X-100) supplemented with protease and phosphatase inhibitor cocktail (Roche) on ice. Samples were then centrifuged at 17,000 g for 30 min at 4°C. Supernatants were collected (soluble fraction) and stored at −80°C.

## In vitro amyloid aggregation assay

Synthetic Aβ40 peptide (rPeptide) was solubilized (1 mg/ml) in 50 mM $NH_4OH$ (pH >11). Upon 15 min incubation at RT and subsequent water bath sonication (5 min), fractions of 100 μg Aβ40 were lyophilized and stored at −20°C. 100 μg Aβ40 was resuspended in 1 ml of 20 mM $NaP_i$, 0.2 mM EDTA, pH 8.0. After water bath sonication, the Aβ40 solution was filtered through an Anatop 0.02 μm filter and stored on ice. Final Aβ40 concentration was assessed from the UV spectrum ([Aβ40] = (A275-A340)/1280 mol/l). Circular Dichroism measurements (Jasco 810 Spectropolarimeter) confirmed the random coil conformation of the Aβ peptide. For the rtQuic aggregation experiments, Aβ40 was diluted to 20 μM final concentration in 20 mM $NaP_i$, 0.2 mM EDTA, pH 8.0 containing 100 μM Thioflavin T. 50 μl of this Aβ40 solution was either mixed with 50 μl SCFA mixture (60 mM total SCFA, i.e. a mixture of Na-acetate, Na-propionate, and Na-butyrate dissolved in equimolar ratios in water containing 100 μM Thioflavin T) or mixed with 50 μl saline solution (60 mM NaCl, 100 μM Thioflavin T) to compare the aggregation of Aβ40 (final concentration 10 μM) in 20 mM NaPi, 0.2 mM EDTA buffer (pH 8.0), either enriched with 30 mM SCFA mixture or with 30 mM NaCl. Experiments were performed in triplicates. Aβ40 solution aliquots were incubated at 37°C in 96-well plates under constant double-orbital shaking in a FLUOStar Omega plate reader (BMG Labtech). The formation of amyloid fibrils was monitored by the Thioflavin T fluorescence. Data were assessed every 15 min (λex = 440 nm), (λem = 480 nm). In control experiments (blanks), 100 μM Thioflavin T was diluted in the 20 mM $NaP_i$, 0.2 mM EDTA, pH 8.0 alone, or enriched with the SCFA mixture (30 mM) or saline solution (30 mM NaCl).

## Immunoblot analysis of mouse brain

For western blot analysis, the soluble fraction from 3 months old APPPS1 mice have been quantified using Bradford assay (Biorad) according to manufacturer's protocol. At least 10 μg per sample have been loaded either on a bis-tris acrylamide (APP, NT, ADAM10, and BACE1) or a Novex 10–20% Tris-Tricine gel (Aβ, C83, and C99) followed by blotting on nitrocellulose membrane (Millipore) using the following antibodies: anti-C-term APP (Y188, 1:1000, Abcam), anti-Aβ (3552, 1:1000, *Yamasaki et al., 2006*), anti-Presenilin 1 (N terminus) (NT, 1:1000, BioLegends), anti-ADAM10 N-terminal (1:1000; R and D Systems), and anti-BACE1 (1:1000, Epitomics). Blots have been developed using horseradish peroxidase-conjugated secondary antibodies (Promega) and the ECL chemiluminescence system (Amersham). An antibody against calnexin (1:1000, Stressgen) has been used as loading control. Three mice per group were analyzed.

Densitometry analysis has been performed using gel analyzer tool on ImageJ (NIH).

## Cell free γ-secretase activity assay

γ-Secretase cleavage assays using C100-His$_6$ as substrate were carried out using purified γ-secretase reconstituted into small unilamellar vesicles (SUV) composed of POPC essentially as described (*Winkler et al., 2012*) except that the reconstitution was performed 0.5× PBS, pH 7.4, instead of 35 mM sodium citrate, pH 6.4. The SCFA mixture (Na-acetate, Na-propionate, and Na-butyrate dissolved in water in equimolar ratios) was added to the assay samples at the indicated final concentrations from stock solutions. Following separation of the assay samples by SDS-PAGE on Tris-Tricine

or Tris-Bicine urea gels (for total Aβ or Aβ species, respectively) (*Winkler et al., 2012*), Aβ was detected by immunoblotting using antibody 2D8 (*Shirotani et al., 2007*).

## Immunohistological microglia analysis

Free floating 30 µm sagittal brain sections were treated for antigen retrieval using sodium citrate buffer and subsequently stained using primary antibodies for Iba1 (1:250, rabbit, Wako) and anti-6E10 (1:1000, Biolegend), and the corresponding secondary antibodies and nuclear counterstain using DAPI. Images were acquired in the frontal cortex using a Zeiss confocal microscope with 40× magnification. Morphological analysis of microglial cells was performed using FIJI software. Each single microglial cell was identified in the Z-stack and a maximum intensity projection (MIP) was created for each individual cell from the image layers covering the individual microglial cell and the resulting image was binarized by thresholding. The area and the perimeter of the cell shape were measured and the CI was calculated (CI = 4π[area]/[perimeter]2) for each cell.

For analysis of ApoE expression in microglia and astrocytes, sections were stained with anti-P2Y12 (1:100, Anaspec) or anti-GFAP (1:200, DakoCytomation Z 0334), mouse anti-6E10 antibody (1:200, abcam), and biotinylated anti-ApoE antibody (1:50). Slides were then labeled with the secondary antibodies and counterstained with DAPI. Samples were imaged using an epifluorescence microscope (40× magnification) and co-localization of Aβ, ApoE, and microglia was analyzed using FIJI software.

## Immunohistological analysis of microglial recruitment

Free floating 30 µm sagittal brain sections have been permeabilized and blocked for 1 hr in PBS/0.5% Triton x-100/5% NGS. Next, samples have been incubated overnight at 4°C with primary antibody anti-CD68 (1:500, AbDserotec), anti-Iba1 (1:500, Synaptic Systems), anti β-amyloid (Aβ) (3552, 1:500, *Yamasaki et al., 2006*), diluted in blocking buffer and stained with the corresponding goat secondary antibody (1:500, Alexa-Fluor, Invitrogen). Immunostainings have been performed on four control mice (APPPS1 supplemented with saline solution) and five treated mice (APPPS1 supplemented with SCFA). Each image has been analyzed through z-stack using Zen 3.2 blue edition (Zeiss) for quantification of microglial recruitment to Aβ plaques. Number of recruited microglia has been normalized on plaque surface (3552 area). Aβ positive microglia (Iba1 positive together with CD68 and 3552 co-localization) have been quantified and normalized on the total number of microglia recruited to Aβ plaques. Histological analysis has been performed by an investigator (AVC) blinded to treatment groups with at least five images per mouse. All data have been normalized to control group.

## Stereotaxic hippocampal injections

All animal handling and surgical procedures for GF as well as the SPF mice were performed under sterile conditions in a microbiological safety cabinet. Animals were anaesthetized, placed in a stereotactic device, and a midline incision was performed to expose the skull. The skull was thinned above the injection point (coordinates: lateral 1 mm, caudal 2.3 mm from bregma) and 2.5 µl mouse brain homogenate with high plaque burden from 8 months old APPPS1 mice or saline were microinjected (Hamilton syringe at 1 µl/min; 2.5 mm depth from the brain surface). The needle was retained for additional 2 min before it was slowly withdrawn. 24 hr after injection, animals were sacrificed, and brain samples processed for immunofluorescence and fluorescence in situ hybridization.

## Single-molecule fluorescence in situ hybridization

Single-molecule fluorescence in situ hybridization (smFISH) was performed using the RNAscope Multiplex Fluorescent Reagent Kit v2 (Advanced Cell Diagnostics) by the manufacturer's protocols. Briefly, free floating 30 µm sagittal brain sections were first dried, washed, and then incubated in RNAscope Hydrogen Peroxide. Antigen retrieval and protease treatment were performed as per protocol. Sections were then incubated with the probe mix (C2-TREM2 and C1-CX3XR1) for 2 hr at 40°C and then immediately washed with wash buffer. Next, sections were incubated with RNAscope Multiplex FL v2 AMP1, AMP2, and AMP3, and then probes were counter stained with TSA Plus Cy3 for C1-*Cx3cr1* and TSA Plus Cy5 for C2-*Trem2*. After washing with PBS and blocking buffer, plaques were identified using primary rat anti-2D8 antibody (1:300, Abcam #16669) and counterstained with

secondary antibody AF488 goat anti-rat (1:200, Invitrogen #A11006). Finally, sections were stained with DAPI. smFISH-stained RNA molecules were counted only within the DAPI staining of the cell; a cell was considered CX3CR1-positive when more than four *Cx3cr1* puncta were present. A cell was quantified as *Trem2* positive when *Trem2* smFISH molecules were observed in the *Cx3cr1*-positive cells.

## Ex vivo Aβ plaque clearance and phagocytosis assay

We investigated microglial phagocytosis by an ex vivo assay as already described (*Colombo et al., 2021*). Briefly, 10 µm brain sections from APPPS1 mice (*Radde et al., 2006*) were incubated on coverslips with anti-Aβ antibodies (6E10, 5 µg/ml) to stimulate microglia recruitment. Primary microglia were isolated from P7 WT mouse pups using the Neural Tissue Dissociation Kit, CD11b microbeads, and a column MACS separation system (Miltenyi Biotec). Purified microglia were resuspended in DMEM F12 with 10% fetal bovine serum and 1% PenStrep (Gibco). Then, isolated microglia were incubated for 5 days at a density of $3 \times 10^5$ cells/coverslip in culturing medium containing 250 µM mixed SCFA (acetate, butyrate, and propionate, each 83.3 µM) or control 250 µM NaCl. Next, sections were fixed and permeabilized and stained with primary antibodies against CD68 (1:500, AbD-serotec) and Aβ (3552, 1:500, *Yamasaki et al., 2006*) and corresponding secondary antibodies as well as the nuclear dye Hoechst and fibrillar Aβ dye Thiazine red (ThR 2 µM, Sigma Aldrich). Full sections were analyzed by tile scan mode on a Leica SP5 II confocal microscope. To evaluate microglial phagocytic capacity, difference in plaque coverage (ThR signal area) was quantified between brain sections incubated with microglia and a consecutive section where no microglia have been plated using ImageJ software (NIH). Each experimental group has been tested in three animals (three independent experiments with two technical replicates each).

## SCFA treatment of primary microglia

Single cell suspension was obtained from P7 WT mouse brains using an enzymatic (200U of papain, Sigma Aldrich) and a mechanical dissociation in DMEM GlutaMAX high glucose and pyruvate (Gibco) supplemented with 10% fetal bovine serum (Sigma). Microglia were isolated using CD11b microbeads and a column MACS separation system (Miltenyi Biotec). Purified microglia were resuspended and plated in DMEM F12 with 10% fetal bovine serum and 1% PenStrep (Gibco) in a 96-well plate format at a density of $5 \times 10^4$ cells/well. At 5DIV culturing medium was replaced with fresh medium containing single SCFA (acetate, butyrate, or propionate, Sigma Aldrich) or an equimolar combination of them (final concentration 250 µM for all conditions). NaCl was used as control in equimolar concentration. After 24 hr incubation, cells were lysed in Qiazol Lysis Reagent and RNA extraction was performed using the RNeasy Mini Kit (all Qiagen).

## Nanostring analysis

RNA samples from primary microglial cell were obtained as stated above. Brain samples were lysed in Qiazol Lysis Reagent and total RNA was extracted using the MaXtract High Density kit with further purification using the RNeasy Mini Kit (all Qiagen). 70 ng of total RNA per sample was then hybridized with reporter and capture probes for nCounter Gene Expression code sets (Mouse Neuroinflammation codeset) according to the manufacturer's instructions (NanoString Technologies). Samples were injected into NanoString cartridge and measurement run was performed according to nCounter SPRINT protocol. Background (negative control) was quantified by code set intrinsic molecular color-coded barcodes lacking the RNA linkage. As positive control code set intrinsic control RNAs were used at increasing concentrations. Data were analyzed using the nSolver software 4.0. In the standard data analysis procedure, all genes counts were normalized to the positive control values and the values for the standard reference genes. Clustering and heatmaps were performed using the ClustVis package on normalized expression values derived from the nSolver report (*Metsalu and Vilo, 2015*). For clustering, the z-scores were calculated using the mean expression of biological replicates per condition and then clustered using K-means. Data were further analyzed using the Ingenuity software (Ingenuity Systems). Differentially expressed genes (with corresponding fold changes and p-values) were used for generating a merged biological network as previously described (*Butovsky et al., 2014*).

## Statistical analysis

All analyses, unless otherwise specified, have been performed using GraphPad Prism software (GraphPad, version 6.0). Sample size was chosen based on experience from previous experiments and exact n numbers are reported in figure legends. The groups containing normally distributed independent data were analyzed using unpaired T-test (for two groups) or ANOVA (for >2 groups). The remaining data were analyzed using the Mann–Whitney U-test (for two groups). Similar variance was assured for all groups, which were statistically compared. A p-value of <0.05 was regarded as statistically significant and displayed in all figures.

## Acknowledgements

The authors would like to thank Kerstin Thuß-Silczak for excellent technical support and Anna Daria for the help in genotyping. The authors are grateful to Mathias Jucker (Hertie Institute for Clinical Brain Research, University of Tübingen, Germany) for providing the APPPS1 mice and David Holtzman (Washington University School of Medicine, St Louis, Missouri, USA) for providing the ApoE antibody.

## Additional information

### Competing interests

Andrew J MacPherson: Reviewing editor, *eLife*. The other authors declare that no competing interests exist.

### Funding

| Funder | Grant reference number | Author |
|---|---|---|
| Vascular Dementia Research Foundation | | Martin Dichgans<br>Arthur Liesz |
| Deutsche Forschungsgemeinschaft | STE847/6-1 | Harald Steiner |
| H2020 European Research Council | ERC-StG 802305 | Arthur Liesz |
| Helmholtz Association | Zukunftsthema "Immunology and Inflammation") | Christian Haass |
| NCL Foundation | | Sabina Tahirovic |
| Alzheimer Forschung Initiative | 18014 | Sabina Tahirovic |
| Deutsche Forschungsgemeinschaft | HA1737/16-1 | Christian Haass |
| Deutsche Forschungsgemeinschaft | EXC 2145 SyNergy - ID 390857198 | Martin Dichgans<br>Christian Haass<br>Arthur Liesz |
| H2020 European Research Council | NMP-2015-686271 | Andrew J MacPherson<br>Mercedes Gomez de Aguero |
| Alzheimer's Association | ADSF-21-831226-C | Sabina Tahirovic |
| Deutsche Forschungsgemeinschaft | LI-2534/7-1 | Arthur Liesz |

The funders had no role in study design, data collection and interpretation, or the decision to submit the work for publication.

### Author contributions

Rebecca Katie Sadler, Validation, Investigation, Visualization, Methodology, Writing - review and editing; Gemma Llovera, Edith Winkler, Frits Kamp, Formal analysis, Investigation, Visualization,

Writing - review and editing; Vikramjeet Singh, Data curation, Formal analysis, Writing - review and editing; Stefan Roth, Data curation, Formal analysis, Investigation, Visualization, Writing - review and editing; Steffanie Heindl, Data curation, Formal analysis, Investigation, Methodology, Writing - review and editing; Laura Sebastian Monasor, Corinne Benakis, Formal analysis, Investigation, Writing - review and editing; Aswin Verhoeven, Data curation, Formal analysis, Visualization, Writing - review and editing; Finn Peters, Formal analysis, Visualization, Methodology, Writing - review and editing; Samira Parhizkar, Investigation, Methodology, Writing - review and editing; Mercedes Gomez de Aguero, Resources, Funding acquisition, Methodology, Project administration, Writing - review and editing; Andrew J MacPherson, Data curation, Supervision, Funding acquisition, Writing - review and editing; Jochen Herms, Martin Dichgans, Supervision, Funding acquisition, Writing - review and editing; Harald Steiner, Formal analysis, Supervision, Investigation, Visualization, Methodology, Writing - review and editing; Martin Giera, Data curation, Formal analysis, Supervision, Funding acquisition, Writing - review and editing; Christian Haass, Resources, Supervision, Funding acquisition, Writing - review and editing; Sabina Tahirovic, Conceptualization, Formal analysis, Supervision, Funding acquisition, Investigation, Visualization, Methodology, Writing - original draft, Writing - review and editing; Arthur Liesz, Conceptualization, Formal analysis, Supervision, Funding acquisition, Visualization, Writing - original draft, Project administration, Writing - review and editing

### Author ORCIDs

Steffanie Heindl http://orcid.org/0000-0003-3576-2702
Laura Sebastian Monasor http://orcid.org/0000-0001-7864-7400
Frits Kamp http://orcid.org/0000-0001-8382-6884
Mercedes Gomez de Aguero http://orcid.org/0000-0002-7132-290X
Harald Steiner http://orcid.org/0000-0003-3935-0318
Martin Giera http://orcid.org/0000-0003-1684-1894
Sabina Tahirovic https://orcid.org/0000-0003-4403-9559
Arthur Liesz https://orcid.org/0000-0002-9069-2594

### Ethics

Animal experimentation: All animal experiments were performed under the institutional guidelines for the use of animals for research and were approved by the governmental ethics committee of Upper Bavaria (Regierungspraesidium Oberbayern). (Regierungspraesidium Oberbayern, license number #160-14).

### Decision letter and Author response

Decision letter https://doi.org/10.7554/eLife.59826.sa1
Author response https://doi.org/10.7554/eLife.59826.sa2

## Additional files

### Supplementary files

- Transparent reporting form

### Data availability

All original Nanostring raw data have been permanently deposited to the NCBI GEO repository: https://www.ncbi.nlm.nih.gov/geo/query/acc.cgi?acc=GSE168036.

The following dataset was generated:

| Author(s) | Year | Dataset title | Dataset URL | Database and Identifier |
|---|---|---|---|---|
| Colombo AV, Sadler RK, Llovera G, Singh V, Roth S, Heindl S, Sebastian Monasor L, Verhoeven A, | 2021 | Microbiota-derived short chain fatty acids modulate microglia and promote A$\beta$ plaque deposition | https://www.ncbi.nlm.nih.gov/geo/query/acc.cgi?acc=GSE168036 | NCBI Gene Expression Omnibus, GSE168036 |

Peters F, Parhizkar S, Kamp F, Gomez de Aguero M, MacPherson AJ, Winkler E, Herms J, Benakis C, Dichgans M, Steiner H, Giera M, Haass C, Tahirovic S, Liesz A

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
