## [Decision Letter]

**Acceptance summary:**

Colombo et al. have re-derived the amyloidosis mouse model (APPS1) under germ-free conditions, leading to decreased plaque load and impaired cognitive performance. They go on to show that short-chain fatty acids are sufficient to increase plaque levels, microglial activation, and ApoE expression, providing a potential mechanism linking dietary intake, gut microbial metabolism, and cognitive performance.

**Decision letter after peer review:**

Thank you for submitting your article "Microbiota-derived short chain fatty acids promote Aβ plaque deposition" for consideration by *eLife*. Your article has been reviewed by 3 peer reviewers, including Peter Turnbaugh as the Reviewing Editor and Reviewer #1, and the evaluation has been overseen by Wendy Garrett as the Senior Editor. The following individual involved in review of your submission has agreed to reveal their identity: John F Cryan (Reviewer #2).

The reviewers have discussed the reviews with one another and the Reviewing Editor has drafted this decision to help you prepare a revised submission.

Summary:

Colombo et al. present an intriguing set of findings from the amyloidosis mouse model (APPS1). Rederivation of this model under germ-free conditions led to both decreased plaque load and impaired cognitive performance. Administration of a cocktail of SCFAs and salt (sodium propionate, butyrate, and acetate) significantly increased plaque levels, microglial activation, and ApoE expression. Together, these findings suggest a potential pathway through which the microbiome could impact cognitive performance. The paper is well-written, with a clear description of the current results and a logical flow to the text and figures. These data are a good starting point for further mechanistic dissection and add another welcome piece to the puzzle of how the microbiota affect the brain.

Essential revisions:

1. Lack of consideration of sex and small sample sizes. Prior work in this area has demonstrated that it is important to look at sufficient numbers of both female and male mice, individually, and not just group them. Moreover, the average number of mice used in each experiment (N=5) is relatively small. We recommend repeating the key experiments with both larger numbers and both male and female mice (analyzed separately).

2. Experimental perturbation of the proposed pathway. The manuscript leads to a nice model; however, the data are descriptive in nature. The impact of this study would be increased substantially if at least one mechanistic link between SCFAs and AB, microglia, or ApoE were experimentally validated. While most of the text avoids making causal claims based on correlative evidence, the one sentence summary states that SCFAs impact disease "via activation of microglial cells and upregulation of ApoE."

3. Identify which SCFA matters. The experiments all rely on a mixture of 3 SCFAs making it impossible to determine which compound is responsible or if there are differences in effects of the individual short-chain fatty acids. There is also high salt in this mixture which confounds the interpretation further. At a minimum, each individual compound needs to be tested using an equimolar amount of salt as a negative control. The authors should also note issues with oral delivery of SCFAs, which does not necessarily mimic production in the colon. Ideally, tributyrin, or a similar ester for acetate or propionate should be used. Another key missing control is the administration of SCFAs to SPF mice. It is also important to be clear that while SCFAs are sufficient to impact AB, there is no evidence in the paper to suggest that they are necessary, the full scope of "key microbial metabolites" remain to be determined. If the authors want to claim necessity, they would need to deplete specific SCFAs in the presence of a complex gut microbiome.

4. Be more cautious in discussing the role of the microbiome in Alzheimer's disease. The background discussion includes studies that show correlations in humans and phenotypic differences in germ-free mouse models, which in our opinion are insufficient to claim a causal role in human disease. The authors should discuss the level of evidence in humans for a causal role of the microbiome and its relative impact relative to other risk factors, including any prospective or intervention studies that have been conducted. They should also take care not to extrapolate differences in intermediate phenotypes in mice (plaque levels, microglial activation, and ApoE expression) to human disease. For example, the one sentence summary says, "contributing to AD disease progression". The authors should also discuss whether or not cognitive performance was evaluated in response to SCFAs.

---

## [Author Response]

Essential revisions:1. Lack of consideration of sex and small sample sizes. Prior work in this area has demonstrated that it is important to look at sufficient numbers of both female and male mice, individually, and not just group them. Moreover, the average number of mice used in each experiment (N=5) is relatively small. We recommend repeating the key experiments with both larger numbers and both male and female mice (analyzed separately).

As suggested, we have repeated the key experiment shown in Figure 1 (i.e. effect of bacterial colonization on plaque load at 5 months of age) and increased the sample size per group to analyze sex-specific effects. Of note, repetition of this experiment has been greatly challenged by the still ongoing pandemic and its impact on our possibilities to breed transgenic mice and maintain germfree mouse colonies. Nevertheless, we were able to increase the sample size per group from 5 to 9 animals (SPF and GF). Additionally, we added a third group of recolonized animals (Rec), this is Ex-GF littermate mice which have been naturally recolonized by co-housing with SPF mice under conventional housing conditions.

Increase of sample size confirmed the substantial effect of bacterial colonization on cerebral amyloid plaque load. The effect of bacterial colonization on plaque load was further corroborated by the increase in recolonized (Rec) mice. The increase of plaque load was statistically only significant in female but not in male mice, however, sample sizes of our experimental groups are still not large enough to study sex-specific differences with sufficient statistical power.

Furthermore, we have increased the sample size for Barnes maze behavior test as well as included the recolonized animal group also for this parameter assessing spatial memory deficits. Increasing the sample size as well as adding the recolonized animal group strengthened the main finding of a significant impact of bacterial colonization on memory deficits.

These additional findings have been added to the revised Figure 1, Supplementary Figure 1 and the respective Results section in the revised manuscript.

2. Experimental perturbation of the proposed pathway. The manuscript leads to a nice model; however, the data are descriptive in nature. The impact of this study would be increased substantially if at least one mechanistic link between SCFAs and AB, microglia, or ApoE were experimentally validated. While most of the text avoids making causal claims based on correlative evidence, the one sentence summary states that SCFAs impact disease "via activation of microglial cells and upregulation of ApoE."

We fully agree that the direct mechanistic link between SCFA, microglia/ApoE and Aβ is still missing. A full mechanistic analysis of the cellular and molecular interactions along this proposed pathway is surely beyond the scope of a single manuscript.

However, in order to strengthen the mechanistic understanding of this newly described SCFA dependent phenomenon, we performed several new experiments:

a) SCFA promote AD progression

A critical claim based on our original findings was that SCFA is a key driver of (microbiota-mediated) AD progression. However, this interpretation was only derived from SCFA-supplemented germfree mice, which are largely devoid of endogenous SCFA. In order to further test the direct disease-modifying properties of SCFA, we treated conventionally housed SPF mice with SCFA in drinking water over 4 weeks. Interestingly, SCFA treated SPF mice had a significantly increased plaque load compared to control-treated SPF mice, unequivocally demonstrating the disease-promoting role of SCFA in this experimental model. This new dataset has been added as the revised Figure 2D.

b) SCFA modulate microglia

We hypothesized that the SCFA modulate microglial function and this effect may contribute to increased Aβ plaque load. This conclusion was mainly based on the morphological and transcriptomic analysis of microglia between GF and SCFA-supplemented GF mice as well as the observed difference in microglial activation towards injected amyloid in SPF and GF mouse brains. In order to further support the impact of SCFA on microglia, we assessed the recruitment of microglia to amyloid plaques and the microglial Aβ content in SPF mice which have been treated with either control or SCFA over 4 weeks in drinking water. This experiment was performed in SPF mice to avoid a potentially counteracting bias of immature microglia in GF mice. Here, we identified that SCFA-treatment increased microglial accumulation at the amyloid plaques but at the same time microglia contained significantly less Aβ. This new dataset has been added as Figure 4F to the revised manuscript.

Next, we aimed to further elaborate on the direct effects of SCFA on microglia. Therefore, we established a primary microglia cell culture system in which we treated microglia with either combined SCFA or single SCFA. We observed a significant upregulation of mRNA transcription of SCFA-treated microglia for various genes previously associated with the activation state of microglia (Casp1, Fcrls, Gadd45, Il6ra, and others). Pathway analysis of the regulated genes suggested mainly a modulatory effect of SCFA on the NF-κB pathway and NF-κB-regulated secretory functions of microglia. Please see the summary of the pathway analysis – further analyses are presented in the new Supplementary Figure to Figure 5.

In addition to performing the new experiments, which all supported a modulatory effect of SCFA on microglia, we have also carefully revised the manuscript to avoid overinterpretation of the findings and toned down our one sentence summary (now denoted as impact statement due to editorial guidelines).

3. Identify which SCFA matters. The experiments all rely on a mixture of 3 SCFAs making it impossible to determine which compound is responsible or if there are differences in effects of the individual short-chain fatty acids. There is also high salt in this mixture which confounds the interpretation further. At a minimum, each individual compound needs to be tested using an equimolar amount of salt as a negative control. The authors should also note issues with oral delivery of SCFAs, which does not necessarily mimic production in the colon. Ideally, tributyrin, or a similar ester for acetate or propionate should be used. Another key missing control is the administration of SCFAs to SPF mice. It is also important to be clear that while SCFAs are sufficient to impact AB, there is no evidence in the paper to suggest that they are necessary, the full scope of "key microbial metabolites" remain to be determined. If the authors want to claim necessity, they would need to deplete specific SCFAs in the presence of a complex gut microbiome.

We agree, that investigating individual SCFA is a relevant aspect which was missed in the original manuscript. In order to address this, we first quantified blood concentrations of the individual SCFA in the portal vein blood of germfree, naturally recolonized (Ex-GF) and SPF mice. Results from this experiment clearly demonstrated that all three SCFA—although at different degrees—are increased in the blood of mice with bacterial colonization. This finding is now included as Figure 2A in the revised manuscript.

The rationale for selecting a combination of all three SCFA was mainly based on the comparability to previous reports that have used the exact same SCFA supplementation protocols in AD (Erny, Nature Neuroscience, 2015) and stroke (Sadler, Journal of Neuroscience, 2020) mouse models. The finding that differences in plasma concentrations (at various degrees) for all three SCFA can be observed between germfree and colonized animals further corroborates this rationale. This issue has now been clarified in the revised Results section in its first paragraph.

Next, we performed the suggested experiment to compare the effect of single SCFA (i.e. Acetate, Butyrate and Propionate) to combined SCFA treatment in equimolar concentrations and in comparison to an equimolar salt control on polarizing microglial activation.

Since it was impossible for us to perform such axenic animal experiments during the current pandemic in vivo using the model of germfree APPPS1 mouse + supplementation (salt control, combined SCFA, Acetate, Butyrate, Propionate), we decided to compare the effects of combined *versus* individual SCFA in cultured primary microglia followed by Nanostring analysis of microglial transcriptomics. This experimental setting also enabled us to validate the potential of SCFA to directly modulate microglia.

This comparison demonstrated that while acetate is the most abundantly detected SCFA in the plasma and also most consistently increased after bacterial colonization (see above), the effect of butyrate and propionate resemble much closer the changes in the transcriptomic profile of microglia treated with combined SCFA than was observed for acetate. Specifically, while we observed a large overlap in the significantly regulated genes between combined SCFA, butyrate and propionate treatment to the respective control, none of the genes detected to be significantly regulated (up- or down-regulated) by acetate overlapped with the combined SCFA treatment. Correspondingly, a comparison by pathway analysis of the differential impact between the treatment groups on biological processes, revealed the same pattern that butyrate and propionate more closely resembled the effects of combined treatment than acetate (see above, answer to essential revision 2). However, each single SCFA treatment only partially resembled the effects of the combined treatment, suggesting a potentially synergistic effect on microglial polarization. These new results are included in the Supplementary Figure to Figure 5.

Additionally, we performed the requested control experiment in which we treated SPF APPPS1 mice with SCFA or control supplementation. As described in more detail above (response to essential revision 2), we observed a significant increase in amyloid plaque load in SCFA-supplemented in comparison to control treated mice.

While we see the relevance of investigating in more depth the production or potentially different effects of SCFA-esters (such as for the proposed tributyrin), we think that this aspect is beyond the scope of the present manuscript. More detailed studies on various aspects of this complex pathway, including the relevance of production and resorption site, will require future studies. However, this aspect has now been included and discussed in the revised Discussion section.

Finally, we clearly want to state that we did not intend to propose that SCFA are necessary mediators for amyloid plaque deposition. We carefully reviewed the manuscript and do not claim that SCFA are “necessary” or “required” for amyloid plaque deposition. However, based on the findings of substantially increased plaque load by the SCFA supplementation, even in the presence of the normal gut microbiome of the SPF APPPS1 mice, we think it is justified to describe SCFA as metabolites that are sufficient to promote Aβ plaque deposition.

4. Be more cautious in discussing the role of the microbiome in Alzheimer's disease. The background discussion includes studies that show correlations in humans and phenotypic differences in germ-free mouse models, which in our opinion are insufficient to claim a causal role in human disease. The authors should discuss the level of evidence in humans for a causal role of the microbiome and its relative impact relative to other risk factors, including any prospective or intervention studies that have been conducted. They should also take care not to extrapolate differences in intermediate phenotypes in mice (plaque levels, microglial activation, and ApoE expression) to human disease. For example, the one sentence summary says, "contributing to AD disease progression". The authors should also discuss whether or not cognitive performance was evaluated in response to SCFAs.

As suggested, we revised and expanded the introduction, Discussion section and one sentence summary in order to not overinterpret our own findings or extrapolate findings from animal studies to human disease.

Specifically, we:

1. Have discussed in detail the very limited available data on the role of microbiota changes in AD patients for a potential disease-modifying function. This discussion has been added to the revised Discussion section.

2. Revised the manuscript text in order to correct for any instance that might have implicated a transfer of findings from animal models to the human disease.

Cognitive performance (spatial memory deficits) were assessed for the key experiments demonstrating the impact of bacterial colonization per se on disease outcome in the APP/PS1 mouse model on SPF, Recolonized and Germfree mice (this data is shown in Figure 1).